# Role of Mast Cells in Human Health and Disease: Controversies and Novel Therapies

**DOI:** 10.3390/ijms26188895

**Published:** 2025-09-12

**Authors:** Miguel Ángel Galván-Morales, Juan Carlos Vizuet-de-Rueda, Josaphat Miguel Montero-Vargas, Luis M. Teran

**Affiliations:** Departamento de Inmunogenética y Alergia, Instituto Nacional de Enfermedades Respiratorias “Ismael Cosío Villegas”, Calzada de Tlalpan 4502, Colonia Sección XVI, Alcaldía Tlalpan, Ciudad de México CP. 14080, Mexico; janvizuet@gmail.com (J.C.V.-d.-R.); jos.miguel.montero@gmail.com (J.M.M.-V.)

**Keywords:** mast cells, C/EBPα, *c-kit*, FcεRIα+, IL-3, IgE, omalizumab, acalabrutinib, remibrutinib, matinib, tyrosine kinase inhibition, mastocytosis, allergic disease, homeostasis

## Abstract

Mast cells have been implicated in allergic diseases such as asthma, rhinitis, conjunctivitis, atopic dermatitis, urticaria, and anaphylaxis. However, it is now well established that they also fulfill critical roles in tissue homeostasis, repair, and defense. Despite considerable progress, their ontogeny, proliferation, and differentiation remain subjects of debate, as does their involvement in a wide spectrum of diseases, including cancer and cardiovascular disorders. What remains indisputable is their essential contribution to both innate and adaptive immune responses. Importantly, the activity of their effector molecules can elicit either protective or deleterious outcomes. A complete absence of mast cells (MCs) in humans would undoubtedly provide valuable insight into their fundamental role in immunity, much as neutropenia and agranulocytosis have historically clarified the functions of neutrophils. In this review, we provide a comprehensive overview of mast cell (MC) biology, emphasizing their functional diversity and pathogenic potential. Furthermore, we highlight emerging therapeutic strategies, particularly the use of inhibitors and monoclonal antibodies, which are reshaping current approaches to conditions such as allergy, mastocytosis, and related disorders.

## 1. Origin and Development of Mast Cells

### 1.1. Mast Cell Precursors Identification

Early studies on MCs focused on the staining properties of their cytoplasmic granules, which are enriched in proteoglycans and proteases. Von Recklinghausen (1863) and Paul Ehrlich (1877) were the first to identify them as distinct immune cells within the granulocyte family.

MCs are widely distributed throughout all tissues, including hollow organs such as the gastrointestinal tract, within blood vessels, beneath epithelial barriers, and in proximity to peripheral nerves. This anatomical localization reflects their central role in antigen recognition and the detection of tissue damage. MCs first appear in the extraembryonic yolk sac on the seventh day of gestation. From there, circulating progenitors migrate into peripheral tissues for complete differentiation and maturation. Embryonic mast cell populations are gradually replaced by definitive progenitors derived from stem cells. Yolk sac-derived MCs originate from mesodermal precursors that form hematopoietic islets in the vitelline region near the allantois, starting around three weeks into gestation [1]. Hematopoietic progenitors subsequently differentiate into multiple lineages, including myeloid, lymphoid, megakaryocytic, and erythroid precursors, which emerge in the fetal liver. Additionally, arteries serve as a secondary organ in hematopoietic formation. At this stage, their fetomaternal interface marker is CD45 low and c-kit/high, with AA4. In the intraembryonic aortogonadomesonephric region, fetal-restricted hematopoietic stem cells are generated until definitive cells appear [2].

In the bone marrow, MC precursors of myeloid origin are identified by the expression of CD34 and CD117 (c-Kit), whereas mature MCs are absent [3]. Immature MCs exit the marrow to be activated by antigens and cytokines for maturation. MCs and basophils share a common progenitor, but basophils leave the marrow in a fully mature state. Hematopoietic stem cells (HSCs) develop into multipotent progenitors (MPPs), which differentiate into common myeloid progenitors (CMPs), and then into granulocyte/monocyte progenitors (GMPs) that differentiate into basophils and MCs. Specification of the mast cell lineage requires the transcription factor CCAAT/enhancer-binding protein alpha (C/EBPα), which is pivotal in determining cellular fate [4,5].

### 1.2. Receptor Expression During Differentiation

The transcription factor C/EBPα is essential for the differentiation of basophil and MC progenitors. Its expression remains low during early differentiation, increases in basophils while decreasing in MCs, as shown in multiple studies [6]. Although the precise nature of this variation in humans remains unclear, the expression levels of C/EBPα and the transcription factor microphthalmia-associated transcription factor (MITF) play a significant role in lineage commitment. Studies in murine models suggest that even minimal levels of C/EBPα are indispensable for mast cell differentiation in humans [7,8]. The classification of the mast cell lineage remains a matter of controversy. Nonetheless, both murine and human MCs require stem cell factor (SCF) signaling for survival and proliferation in culture. Distinct progenitor populations are identified by markers such as Lin^−^, c-Kit^+^, CD34^+^, Sca-1^low^, Ly6c^+^, FcεRIα^+^, CD27^+^, β7^−^, and Flk-2^−^, and at least ten molecular signatures have been associated with lineage-specific transitions [9,10]. Furthermore, the groundbreaking research conducted by Arinobu et al. in 2005 [11] revealed that single cells within the granulocyte/monocyte progenitor can differentiate into neutrophils, basophils, and mast cells. This population is defined as Lin− IL-7Rα− c-Kit+ Sca-1− CD34+ FcγRII/III^hi^, underscoring the complexity of lineage plasticity [11,12]. Nevertheless, additional evidence suggests that primary receptor sequencing is more common.

Phenotypic heterogeneity among mast cells reflects the influence of local tissue environments. It is essential to note that mast cells are likely derived from multipotent progenitor cells (MPPs) and can be identified by the presence of the high-affinity IgE receptor (FcɛRI), which can be expressed as either FcɛRI-negative or FcɛRI-positive cells. Mast cell in the periphery (MCp), originating from the bone marrow, can be detected in the blood of BALB/c and C57BL strains of mice, existing as both FcɛRI− and FcɛRI+ cells. In peripheral tissues, the MCp in C57BL mice tends to express low levels of FcɛRI+ [13]. Therefore, it can be concluded that some MCp likely originate from MPPs, bypassing the other lineages (see Figure 1) [14].

### 1.3. Receptors for Maturation and Tissue Migration

Immature mast cells circulate in the blood expressing c-Kit, IL-3R, and FcɛRI+. Their migration is tissue-specific, and although they do not divide, they can migrate massively when necessary [15]. Different tissues require specific membrane molecules for migration. For instance, migration to the intestine requires the integrin α4β7 and the CXCR2 chemokine receptor in MCp [16]. Additionally, vascular mucosal cell adhesion molecule (MAdCAM-1) and vascular cell adhesion molecule 1 (VCAM-1) are expressed in the intestinal endothelium [17,18]. In allergic pulmonary inflammation, the recruitment of MCs also depends on α4β7 and α4β1 integrins, with VCAM-1 expression regulated by CXCR2, and CCR2/CCL2 pathways in the pulmonary vascular endothelium [19,20,21]. Blockade experiments have confirmed this dependency. Moreover, peritoneal MCs express αIIbβ3 integrin (RGD type) and glycoprotein IIb [22].

The expression of surface proteins during MCs migration is influenced by soluble growth factors, such as Vascular Endothelial Growth Factor (VEGF) and Endothelial Growth Factor (EGF), which modulate their movement, adhesion, rolling, apoptosis, and differentiation [23,24]. In vitro studies of chemotaxis suggest that other mediators are also involved. For example, prostaglandin receptors EP2, EP3, and EP4 in peritoneal MCs cooperate in this process [25]. In the skin, adhesion, migration, and the expression of surface proteins on MCs are similarly influenced by VEGF, EGF, and chemokine CCL2 binding ot its receptor, CCR2 [26,27]. Additionally, laminins α3 and α5 promote the migration and invasion of MCs in the epidermis via integrin α3β1 [28].

Human CD34+ derived MCs express multiple 5-hydroxytryptamine (5-HT) receptors, which enhance adhesion and migration in both human and murine cells [29]. These MCs express the high-affinity IgE receptor (FcεRI), which they acquire in the bone marrow. Their activation is primarily triggered when IgE binds to a specific antigen, leading to the release of several mediators, including preformed substances such as histamine, tryptase, chymase, carboxypeptidase, cathepsin G, serotonin, heparin, major basic protein, acid hydrolases, peroxides, and phospholipase. Additionally, lipid mediators such as leukotrienes and prostaglandins (LTB4, LTC4, PGE2, PGD2, PAF), along with cytokines (IL1, IL5, IL6, IL6, IL13, IL16, IL18, TNFα, TNFβ, IFN-α, IFN-β) and chemokines: IL8(CXCL8), MCP-1 (CCL2), MCP-3(CCL7), MIP-1α(CCL3), MIP1β (CCL4), RANTES (CCL5), eotaxin (CCL11), contribute to this process. Growth factors like SCF, M-CSF, GM-CSF, bFGF, VEGF, NGF, and PDGF are also involved. Notably, IgE can promote mast cell survival and increase FcεRI expression even in the absence of an antigen, suggesting a self-activating role [30]. In addition, IgE-independent activation pathways are increasingly recognized as critical contributors to disease, although these mechanisms remain poorly understood (see Figure 2).

MC can release IL-1β and other members of the IL-1 family, including inflammatory and anti-inflammatory cytokines. Although they are not the main source of IL-1β, mast cells can produce and release it in response to certain stimuli. TNFα, IL-1β, and IL-6 are the main pro-inflammatory cytokines. MC, together with T cells and macrophages, releases interleukin-10 (IL-10), a pleiotropic immunoregulatory cytokine with multiple biological effects. IL-10 inhibits Th1 inflammatory cells, particularly TNF generated mainly by macrophages and MCs, and downregulates IFN-γ, IL-1, and IL-6. IL-37 is a cytokine from the same family that binds to the α chain of the IL-18 receptor and inhibits inflammatory mediators, including TNF, IL-1, IL-6, IL-33, and nitric oxide (NO). IL-10, IL-37, and IL-38 are the main anti-inflammatory agents [31,32].

## 2. Activation Mechanisms of Mast Cells

### 2.1. IgE/Antigen-Dependent Activation

MCs regulate inflammation and immunity by maintaining tissue homeostasis while responding to danger signals, releasing pro-inflammatory mediators either immediately or in a delayed manner depending on the stimulus [33]. Their main immunomodulatory function relies on IgE binding to the high-affinity FcεRI receptor, the central mechanism of mast cell activation [34].

FcεRI activation occurs through three main mechanisms. First, FcεRI, IL-3, and SCF promote mast cell migration independently of IgE. Second, type 2 immune responses, driven by Th2 lymphocytes secreting IL-4, IL-5, and IL-10, enhance IgE production and eosinophilia. Third, the antigen–IgE–FcεRI complex regulates type I hypersensitivity and late-phase type IV allergic reactions (Figure 2) [35]. This process involves the release of preformed mediators triggered by the binding of IgE to antigen via FcεRI [36]. These mechanisms can currently be inhibited with IgG1 monoclonal antibodies such as ligelizumab (QGE031) and omalizumab (Xolair^®^) [37]. The intensity of the response varies depending on the antigen concentration, potentially resulting in local or systemic IgE-mediated anaphylaxis [38,39]. Studies suggest that IgE enhances MCs survival by promoting FcεRI expression, which leads to increased numbers and viability of MCs. This activation occurs through MAPKs signaling pathways (ERK/JNK/p38) [40]. In transplantation experiments with IgE-producing hybridoma in mice, this effect is confirmed, resulting in a rise in gastric MCs following MAPK activation [41]. Conversely, inhibitory mechanisms, such as the activation of the immunoreceptor tyrosine-based inhibitory motif (ITIM) in MCs or antibody binding to the mast cell function-associated antigen (MAFA), diminish FcεRI signaling, thus counteracting its anti-apoptotic effects (Figure 3) [42].

Activation of the MAPK pathway is central to many pro-inflammatory activities of immune cells. In macrophages, lymphocytes, and MCs, MAPKs such as p38, ERK, and JNK are triggered by diverse receptors to regulate inflammation, proliferation, differentiation, and apoptosis. Pathogens, including viruses, bacteria, and parasites, stimulate this pathway via integrins (LFA-1/αLβ2 and Mac-1/αMβ2) on MCs, inducing cytokine release and activating NF-κB through MAPKs. This cascade can culminate in a cytokine storm, primarily mediated by IL-1, TNF, and IL-6. In addition, downstream kinases, such as kinase 1, activate transcription factors that drive the production of inflammatory cytokines [43].

### 2.2. Inappropriate Mast Cell Reaction or Response

MCs and basophils can misinterpret harmless antigens as threats, initiating an IgE/FcεRI-mediated allergic response. Presentation of such antigens by antigen-presenting cells (APC) or B lymphocytes induces IgE production, which binds FcεRI on MCs and triggers degranulation. The release of preformed mediators and pro-inflammatory lipids amplifies inflammation, with symptom severity proportional to mediator load [44].

Both antigen-dependent and antigen-independent pathways, along with FcεRI surface expression, are crucial in determining the outcome of allergic responses [45]. Although IgE can also bind FcγRII, FcγRIII, and galectin-3 on certain mast cells, its interaction with FcεRI is the principal determinant of mast cell function [46]. Studies in mice show that IgE binding to the FcεRI β-chain enhances mast cell responses, whereas certain variant chains can attenuate this effect [47,48]. Dysregulation is associated with multiple conditions, including allergic rhinitis, asthma, atopic dermatitis, food and drug allergies, rheumatoid arthritis, contact dermatitis, hay fever, urticaria, and allergic proctocolitis [49,50,51,52,53,54]. Individuals with these diseases may experience an exaggerated response, such as anaphylaxis or hyperreactivity, upon exposure to the allergen.

### 2.3. Non-IgE-Mediated Mast Cell Activation

MCs play an essential role in allergic reactions and the immune response. While the late phase of type I hypersensitivity (IgE-mediated allergic reaction) is a significant trigger for mast cell degranulation, it is not the only factor that contributes to this process. Various molecules can activate MCs, leading to the release of different mediators and triggering inflammatory or allergic responses. These stimuli can cause not only degranulation but also the selective release of mediators, as well as influence the proliferation, differentiation, and migration of various cells. Some stimuli act synergistically with IgE-mediated activation [55]. This broad reactivity underscores the versatility of MCs in different physiological processes. Table 1 lists the mast cell-activating molecules and their subsequent effects.

## 3. Protective Roles in Homeostasis and Defense

### 3.1. Mast Cells as Mediators of Health

Mast cells regulate the epidermal barrier, control vascular permeability, and modulate keratinocyte activity, even in the presence of commensal microbiota. Beyond their role in skin integrity, MCs are essential for wound healing, thermoregulation, UV protection, and pregnancy physiology [94,95,96]. This functionality largely stems from their release of mediators, such as histamine, VEGF, IL-6, and IL-8, which increase vascular permeability and facilitate vasodilation, thereby promoting the migration of inflammatory cells for tissue repair. In this process, MCs activate key cells, including fibroblasts and keratinocytes [97]. During scar formation, the release of IL-4, VEGF, bFGF, FGF-2, PDGF, TGF-β, and NGF by mast cells promotes fibroblast proliferation, thereby aiding in the development of scar tissue [98].

### 3.2. Proteases in Tissue Remodeling

In humans and mice, there are two main types of MCs: tryptase-positive mast cells (MCT) and tryptase and chymase-positive mast cells (MCTCs), characterized by a serine protease signal. The distribution of these subtypes differs as follows: MCTs are found in mucosal areas in both humans and rodents, while MCTCs resemble rodent connective tissue MC. Upon activation, mast cells degranulate, releasing not only proteases but also various bioactive molecules such as heparin, tryptase, chymase, and active matrix metalloproteinases [99]. This enzyme cocktail promotes the degradation and remodeling of the extracellular matrix, which is further supported by the de novo proliferation of tissue cells, as well as the synthesis of elastin and collagen. Key mitogenic and angiogenic factors, such as basic fibroblast growth factor (bFGF) and vascular endothelial growth factor (VEGF-A), also play a role in skin remodeling during both healing and natural development. MCs communicate with adjacent fibroblasts (FB), playing an essential role in skin health through both stimulated and constitutive mediator release [100].

### 3.3. Angiogenesis and Wound Repair

De novo angiogenesis, driven by mesenchymal stem cells (MSCs), is essential for tissue development and growth. This complex process involves the breakdown of the extracellular matrix, modulation of cell interactions, and the proliferation and differentiation of endothelial cells [101]. Although crucial for normal growth, angiogenesis is typically undetectable in healthy adult tissues due to the balance of angiogenic and anti-angiogenic factors, as well as the regulation of cell recruitment and migration. Key MCs regulatory factors for angiogenesis include IL-4, VEGF, bFGF, PDGF, TGF-β, and FGF-2-induced fibroblast growth [102]. Angiogenesis is most pronounced in hormone-dependent adult tissues, such as the uterus and placenta, during gestation. In adults, it is primarily linked to hypoxia, inflammation, and tissue repair.

### 3.4. Reproductive System Regulation

MCs maintain homeostasis through interactions with both immune and non-immune cells, which is essential for barrier function and tissue stability. This highlights their key role in reproduction and the menstrual cycle, as observed in murine models, suggesting a physiological function that remains partially understood [103]. Moreover, MC activation shows variations that depend on the estrous cycle and hormonal status in the female reproductive systems of humans, rodents, and canids, especially around ovulation and during endometrial preparation for implantation [104].

During the premenstrual phase, the activity of MCs and their matrix-degrading enzymes increases, peaking during menstruation [105]. Hormonal changes, particularly the pregnancy-related peaks in estrogen and progesterone, influence the migration, maturation, and activation of MCs in mice [106]. In both humans and mice, the pregnant uterus contains a significant population of MCs, including trophoblastic MCT and decidua MCTC, which display different levels of differentiation and reparative functions [107]. The degranulation of MCs is essential for the development of the decidua, vascularization, cell proliferation, endometrial and placental growth, and may also play a role in normal fetal development [108]. After delivery, mast cell proteases contribute to endometrial remodeling, the reorganization of uterine tissue, and the de novo revascularization of the uterus [109].

### 3.5. UV Radiation Protection

Histamine, produced by MCs, functions as a potential immunosuppressant. While its overexpression can cause contact hypersensitivity and tissue damage, inhibiting it may protect against ultraviolet B (UV-B) radiation, without affecting keratinocytes. Histamine prevents allergic contact dermatitis (caused by sources such as poison ivy) and skin damage from chronic low-dose UV-B radiation [110,111]. Traditionally, MCs were viewed as amplifiers of inflammation through the release of pro-inflammatory mediators; however, recent studies suggest that besides histamine, MC-secreted IL-10 may limit leukocyte recruitment and reduce tissue injury in skin inflammation [112]. These findings underscore the crucial role of MCs in the immune response, as they interact with dendritic cells and B and T lymphocytes, thereby modulating the intensity of the immune response. Although few in vivo studies have focused on their anti-inflammatory functions, mediators such as TGF-β, IL-4, and IL-10, secreted by MCs, exhibit immunosuppressive properties, negatively regulating responses in specific tissues [113].

A central regulator is STAT5, activated by FcεRI, IL-3, and SCF, which supports early mast cell homeostasis. Control of the JAK–STAT pathway is further modulated by suppressors of cytokine signaling (SOCS) and protein inhibitors of activated STAT (PIAS), both of which act as negative regulators. Deeper insight into MC biology and JAK–STAT regulation may prove pivotal for developing novel therapeutic strategies [114].

### 3.6. Maintenance of Tissue Homeostasis

The homeostasis and intercellular communication of MCs are essential for their protective role against pathogens, supporting immune surveillance, pathogen clearance, and antimicrobial defense. MCs are strategically situated in areas prone to infection, including skin, mucosa, lymphatic drainage, gut, and airways. However, their functionality relies on intercellular communication mediated by different molecules, which modulate responses and ensure the survival of affected cells. Histamine and proteases, key mediators stored in granules, play a crucial role in influencing endothelial activation and vascular permeability. To enhance immune responses to viral infections several pathways can be activated: (1) release of pro-inflammatory cytokines (TNF, IL-1β, IL-6, IL-8); (2) activation of Toll-like receptors and G-protein coupled receptors (e.g., MRGPRX2); (3) stimulation of chemokine receptors (CCL3, CCL4, CCL5); and (4) modulation of Th1 and Th2 cytokine responses [115] (Figure 3). In both rats and humans, MCs communicate intracellularly by releasing (4) interleukins, leukotrienes, MCP-1, and prostaglandins (PGs), which (5) enhance the expression of the B-cell receptor (BCR), chemokine ligand (CCL5) and receptor (CCR5), macrophage receptors, CXCL1/CXCL2, and the T-cell receptor (TCR) to activate B cells, T cells, and macrophages [78]. Additionally, MCs interact with diverse cell types, including smooth muscle, epithelial, endothelial, skin, adipocytes, nerve, neutrophils, lymphocytes, macrophages, and DCs. For complete receptor profiling in mast cell signaling, see the reference [116,117,118,119,120,121,122,123,124,125,126,127,128].

## 4. Mast Cells in Infections

### Combating Infections

MCs act as sentinels against pathogens, tissue damage, and cellular stress [129]. Their role in tissue repair and innate immunity is critical, significantly affecting the repair and induction of adaptive immunity [130]. MCs influence T cells directly and indirectly by modulating antigen-presenting cells (APCs), including DCs, B cells, and macrophages. Activation of MCs begins with PAMPsand DAMPsvia PRRs, which can trigger the complement system or cytokine release through interactions with Fc receptors and other molecules [131].

MCs release molecules extracellularly or intracellularly, enabling them to combat bacteria through complement and PRRs, such as TLRs [132,133]. Key bacterial components include lipopolysaccharide (LPS), heat shock proteins, and flagellin, which are found in the bacterial wall. Bacterial infections typically activate MCs via TLR4 and TLR2, connecting innate and adaptive immune responses to DC activation [134,135]. The absence of DCs during infections leads to delayed healing due to reduced proinflammatory cytokine secretion by MCs, as shown in studies with *W/Ww* Kit knockout mice, where healing was absent [136,137,138]. Additionally, bacterial infections stimulate both mast cells and neurons through MRGPRX2 receptors, which are decisive for cutaneous immunity and nociception (Figure 3) [139,140,141].

The response to respiratory viral infections, particularly respiratory syncytial virus (RSV), is well-documented to lead to the development of hypersensitivity and asthma in affected individuals [142]. This altered hypersensitivity is linked to the RSV G protein, a glycoprotein that resembles the CX3CL1 ligand, activating MC via the CX3CR1 receptor [143,144]. Additionally, mosquito-borne viruses like dengue provoke a Th1 response, leading to endothelial damage and vascular inflammation, which also activates mast cells. Moreover, components of mosquito saliva can induce allergic reactions. The virus’s entry through the skin into the vascular system significantly influences the responses of MCs by releasing mediators in the vascular endothelium and skin [145,146,147,148].

The MCs play a well-established role in immune response against parasitic infections, particularly in cutaneous infections. Classic examples include *Trypanosoma*, *Plasmodium*, and *Leishmania* spp. [149]. In Chagas disease, caused by *Trypanosoma cruzi*, significant damage occurs to the connective tissue where MCs are located, leading to scarring and remodeling. MCs proteases, such as tryptase and chymase, modulate the activation of inflammatory cells. Increased numbers of tryptase-immunoreactive MCs have been observed in the esophagus of infected patients, contributing to megaesophagus [150]. Clinical studies indicate that nearly all malaria patients experience notable MCs degranulation in the skin, which serves as a strong indicator of parasitemia and disease severity [151]. While *Leishmania* spp. primarily target macrophages in cutaneous infections, it has been demonstrated that *L. major* and *L. infantum* can also infect MCs. In these infections, activated MCs play a role in recruiting DCs, which can directly eliminate parasites [152].

The responses of MCs to different fungi are diverse. For yeasts, mouse peritoneal MCs show activation and release early proinflammatory cytokines such as IL-1β, IL-6, IL-8, and TNF-α without degranulation upon contact with conidia [153]. In contrast, *Aspergillus fumigatus* triggers MCs degranulation and the release of proteases through FcεRI/IgE/antigen receptor cross-linking, accompanied by the production of preformed mediators and leukotrienes (Figure 3) [154,155]. This pathogen also promotes allergic lung disease, characterized by elevated levels of IgE [156]. Other fungi, such as *Candida albicans* and *Paracoccidioides* spp., are managed by MC surveillance and degranulation, resulting in the secretion of TNF-α, IL-6, IL-10, CCL2, CCL4, and NOS [155,157,158].

## 5. Mast Cell Dysfunction and Disease

Mast cell dysfunction encompasses disorders in which MCs either proliferate excessively or release abnormal amounts of inflammatory mediators. This may result from the uncontrolled expansion of abnormal cells or hyperresponsiveness to triggers, leading to the overproduction of mediators. Clinical manifestations range from mild to severe, with the most recognized entities being mastocytosis, mast cell activation syndrome (MCAS), and hereditary alpha tryptasemia (HAT) [159], summarized in Table 2.

### 5.1. Mastocytosis

Mastocytosis is classified into three main types: cutaneous, systemic, and mastocytic leukemia. It is characterized by the clonal proliferation of MCs in one or more organs, complicating diagnosis due to significant clinical heterogeneity [159]. Cutaneous mastocytosis, most commonly seen in children, is limited to the skin and typically resolves by adolescence. In contrast, adult-onset cases are rare and may indicate a more severe condition [160]. Systemic mastocytosis involves the infiltration of multiple organs and can present as indolent, bone marrow mastocytosis, or latent systemic mastocytosis [161].

More severe forms of mastocytosis include aggressive systemic mastocytosis, systemic mastocytosis with associated myeloid neoplasm (previously known as systemic mastocytosis with associated hematological neoplasm), and mastocytic leukemia. Until now, the most common cases have been in patients with indolent forms of mastocytosis who are diagnosed after experiencing severe anaphylaxis to hymenoptera venom. Hymenoptera allergy is an exaggerated immune system reaction (anaphylaxis) to insect stings, such as those from bees, wasps, and other members of the Hymenoptera order. This reaction can be fatal and also includes symptoms such as hives, swelling, and difficulty breathing [162]. There is also mast cell sarcoma, a rare and aggressive variant that may affect the skin and other organs (extracutaneous mastocytoma) [163].

In summary, systemic mastocytosis is a multi-organ disease with a broad clinical spectrum, ranging from asymptomatic cases to severe neoplasms (see Table 3) [164,165]. These conditions primarily affect adults, and their management requires analysis of both abdominal and skeletal bone marrow [166]. Treatment is based on bezuclastinib and elenestinib, which is a type 1 tyrosine kinase inhibitor (TKI) that acts by specifically blocking the KIT D816V mutation. This mutation leads to the uncontrolled proliferation of mast cells, resulting in a range of symptoms and organ damage. By inhibiting the mutated KIT protein, bezuclastinib aims to reduce the mast cell burden and alleviate the symptoms associated with systemic mastocytosis [167,168]. The classification of mastocytosis is based on genetic criteria, particularly mutations in the *c-kit* gene, which alter the function of the *c-kit* protein. The first researchers to link mast cell neoplasia with specific genetic mutations identified two point mutations in *c-kit*: valine to asparagine at amino acid 816 and glycine to valine at amino acid 560 in a human mast cell leukemia cell line [169,170]. Numerous studies confirm that *c-kit* mutations are the most frequent in the development of these neoplasms [171,172,173].

### 5.2. Mast Cell Activation Syndrome (MCAS)

MCAS is a chronic, multisystem disease characterized by the abnormal release of mast cell mediators, resulting in allergic and inflammatory reactions across multiple systems. Diagnosing MCAS is complex, and it predominantly affects women, particularly during pregnancy, postpartum, and lactation, with risks such as miscarriage and postpartum hemorrhage. Research focused on pregnant and postpartum women remains limited, despite an estimated prevalence of around 17%, highlighting its potential to cause significant morbidity [174]. Therefore, obstetricians and gynecologists must be vigilant about MCAS and its effects during pregnancy, delivery, and the postpartum period. The *c-kit* mutation, especially the *KIT* D816V mutation, serves as a consistent marker for the syndrome [175,176].

MCAS is not classified as a subtype of mastocytosis or a premalignant condition. However, patients with systemic mastocytosis experience higher rates of MCAS and severe anaphylactic reactions compared to healthy individuals. MCAS can be categorized into different variants (Table 2) [177,178].

### 5.3. Allergic Diseases

Tissue-resident MCs are primarily found near external barriers such as the skin and the mucosal surfaces of the lungs, gastrointestinal tract, conjunctiva, and nose. This strategic placement enables them to initiate an inflammatory response and contribute to tissue repair rapidly. Their involvement in type 2 inflammatory disorders, particularly allergies, has been a focus of research, particularly in relation to respiratory conditions such as asthma and rhinitis. MCs contribute to the severity and development of these Th2 diseases through the biosynthesis of eicosanoids, the release of cytokines, and the secretion of preformed mediators, including histamine and proteases [179].

Allergic diseases arise in response to allergens and include conditions like allergic asthma, conjunctivitis, rhinitis, oral pollen allergy syndrome (OPAS), nasal polyposis, atopic dermatitis, urticaria, hay fever, food and drug allergies, cold and sun allergies, pet and latex allergies, as well as airway hyperresponsiveness (AHR) [180]. The respiratory inflammatory response is characterized by epithelial secretions, activated eosinophils, MCs, and T and B lymphocytes found in bronchial biopsies and bronchoalveolar lavage fluid (BALF) of individuals with asthma and related respiratory allergies [181].

Allergic diseases are characterized by elevated allergen-specific IgE levels (type 1 hypersensitivity) and a Th2 cytokine profile. MCs activation releases mediators such as histamine, proteases, and leukotrienes, causing bronchoconstriction, vasodilation, and increased blood flow, which leads to irritation, increased venular permeability, and mucus secretion [182]. This buildup can result in generalized rashes, urticaria, and airway inflammation, potentially escalating to anaphylaxis [183]. The primary allergy response occurs when IgE binds to the high-affinity receptor FcɛRI on mast cells, basophils, and the gastrointestinal mucosa. However, IgE can also bind with lower affinity to FcεRII (CD23) and the εBP-binding protein (galectin-3) [184]. Certain ligands, such as quinolones like ciprofloxacin, vancomycin, morphine, and rocuronium, can provoke allergic-like symptoms without IgE involvement by binding to the MC-specific receptor MRGPRX2.

Additionally, MCs can release mediators selectively, enhancing the secretion of other neuro-responsive mediators [185,186]. The current treatment is based on either omalizumab or Ligelizumab, which binds to IgE with a higher affinity than omalizumab and has greater potency in inhibiting IgE/FcεRI signaling, thereby reducing mediators in the area. Omalizumab is more potent in inhibiting IgE binding to CD23. In fact, the current treatment is based on allergen avoidance and mast cell mediator antagonists [187].

### 5.4. Urticaria

Urticaria is primarily categorized into acute and chronic forms. Chronic urticaria is further divided into chronic spontaneous urticaria (CSU) and chronic induced urticaria (CIndU). Both types arise from activation and degranulation of MCs and basophils, leading to the release of histamine and other mediators. This common itchy skin condition can manifest as either small bumps or large welts [188]. Chronic urticaria can persist for over six weeks, frequently without an identifiable trigger. CSU occurs independently of external factors, while CIndU is triggered by specific stimuli, such as temperature changes or rare conditions like vasculitis, which may require a biopsy for diagnosis. The etiology of chronic urticaria is complex, leading to discussions on its classification as either an autoimmune disorder or a tissue-resident condition. Factors such as psychological stress and nonsteroidal anti-inflammatory drugs (NSAIDs) can exacerbate CSU symptoms, so managing these triggers may alleviate discomfort [189]. Omalizumab, originally used to treat asthma, is still used to treat chronic spontaneous urticaria that does not respond to other treatments. An antibody that blocks the action of IgE is effective [190]. Several products are currently in phase 1 trials to treat this problem and are named EVO756, Ak006, INCB000262, EP262, IMM47, and others, with very promising results (see Figure 2). Both forms demonstrate varying degrees of MC degranulation, which can be instigated by: (1) activation of MC receptors by environmental stimuli; (2) upregulation of specific receptors; and (3) intracellular dysregulation linked to overexpression of tyrosine kinase SYK immunoreceptor tyrosine-based activating motif (ITAM-related) or underactivation of inhibitory pathways involving inositol phosphatases containing Src homology 2 (ITIM-related) [191,192]. Bruton’s tyrosine kinase inhibitors act specifically downstream of the enzyme’s action, preventing mast cell activation. Acalabrutinib and remibrutinib block the action of the abnormal or mutated protein that signals cells to multiply and activate; both inhibitors are used for the treatment of chronic urticarial [193].

Various stimuli activate sensory neurons, but the sensation of itching arises from mediators that activate pruriceptors in sensory nerve fibers. These fibers transmit signals from the skin to the spinal cord and brain via the dorsal root and trigeminal ganglia. The brain processes and interprets these signals as itching, leading to a response that triggers scratching to alleviate the sensation [194]. Pruriceptor activation can be histaminergic or non-histaminergic. In allergic conditions like urticaria, histamine and proteases from MC granules are primary mediators of itch.

Additionally, cysteine proteinase can activate proteinase-activated receptors 2 (PAR2) and PAR4 [195], with PAR2 being particularly relevant in itch, as it involves multiple afferent pathways within the peripheral nervous system. Non-histaminergic itch occurs when skin is irritated, prompting keratinocytes and local immune cells to release mediators that stimulate pruritic sensory neurons in the dermis and epidermis. Mediators such as TSLP and IL-33, primarily secreted by keratinocytes, along with cytokines like IL-4, IL-13, and IL-31, are associated with Th2-type T-helper lymphocytes and DCs, playing a role in this process. The ineffectiveness of antihistamines in many cases of chronic pruritus suggests that histamine is not the sole mediator, highlighting the limited understanding of the causes of chronic pruritus [196]. Various theories exist regarding the pathogenesis of CSU, including MCs and basophil activation, possibly influenced by IgG and IgE autoantibodies, suggesting an autoimmune etiology. Degranulation of mast cells and basophils, triggered by allergens, toxins, infections, and autoantibodies, activates the membrane receptors of these cells. In addition, eosinophils may also contribute to mast cell activation in CSU, though their precise role is not yet fully elucidated [197]. Acute urticaria is associated with the release of inflammatory mediators, involving an IgE-mediated type 1 hypersensitivity reaction, with symptoms appearing minutes after exposure to allergens such as certain foods, insect venom, or medications. In contrast, chronic urticaria encompasses a range of activation pathways that may overlap. When autoimmunity is suspected, some cases are classified as idiopathic urticaria, often associated with autoimmunity, activation of the coagulation pathway, neuroimmune dysregulation, or infections that lead to IgG production [198]. MC activation occurs via three pathways regulated by receptors with increased expression. As outlined in Table 4, key mediators in this activation include neurotrophins, neurotensin, substance P, complement proteins, free IgG, and factors from the coagulation cascade. Building on this concept of the involvement of a broader range of receptors, antibodies such as barzolvolimab (CDX-0159) and briquilimab, humanized antibodies that inhibit KIT activation (see Figure 2), are under investigation for the treatment of chronic urticaria [199].

IL-4 plays a crucial role in the immune response by promoting the differentiation of naive T cells into Th2 cells, enhancing IgE production by B cells, and inhibiting Th1-mediated inflammatory responses. It increases IgE production by B lymphocytes and fuels the allergic cycle. IL-31 is a cytokine produced mainly by Th2 cells and MCs. It is a crucial molecule involved in the sensation of itching in urticaria and atopic dermatitis. Research into inhibition therapy is currently underway. In mastocytosis, IL-31 is directly associated with the activation of the IL-31R receptor in neurons, increasing the intense itching experienced by patients with these diseases. In mastocytosis, the dysregulated production of IL-4, IL-31, and other cytokines exacerbates the disease’s severity and perpetuates a cycle of mast cell activation, immune response, and symptom manifestation [200].

### 5.5. Mast Cells in Cardiovascular Disease

Mast cells are located in the heart and large vessels, primarily within the myocardium, pericardium, aortic valve, and near the vagus nerve and branches of the bundle of His. They release a range of vasoactive and proinflammatory mediators, influencing angiogenesis, lymphangiogenesis, tissue remodeling, and fibrosis. Their secretion includes preformed mediators, such as histamine, tryptase, and chymase, as well as de novo synthesized mediators, including leukotrienes, prostaglandins, cytokines, and chemokines, which activate various immune cells, such as neutrophils and macrophages. The transcriptional profiles of different MC populations suggest heterogeneity in gene and protein expression, enabling them to perform distinct, sometimes opposing functions in response to changes in the tissue microenvironment. Human cardiac MCs differ significantly from those found in other organs, highlighting their potential dual role either promoting or protecting against cardiovascular disease [201].

Cardiac mast cells (CMCs) have gained attention due to their diverse functions in cardiac repair and damage prevention. Their strategic localization in coronary vessels, between myocytes, in the epicardium, near cardiomyocytes, and sensory neurons containing substance P, as well as in atherosclerotic plaques, suggests a critical role in inflammation, remodeling, and cardiac homeostasis. Activation of CMCs, through IgE cross-linking at its high-affinity receptor or via complement pathways (C3a and C5a) during cardiac or systemic anaphylaxis, can influence tissue damage. In a rat model of myocardial ischemia, complement depletion following CMCs activation was associated with reduced tissue injury, underscoring the essential role that CMCs play in maintaining the heart’s balance [202].

Recent research has linked MCs to the development of atherosclerosis, a chronic inflammatory disease. Initially, atherosclerotic lesions form lipid plaques and cellular debris on the arterial wall, which can progress to severe atherothrombotic complications. This disease primarily affects the intima, the inner layer of the arterial wall, while the muscular and adventitial layers remain relatively intact. Although CMCs are concentrated around the microvessels, they have not been implicated in this process, as the avascular areas lack these cells [203]. In contrast, the highly vascularized adventitia contains a dense population of mast cells. As atherosclerotic lesions develop, the intimal tissue often becomes hypoxic. Mast cells facilitate this process by promoting the growth of microvascular branches from the adventitial vasa vasorum into the intima. They release angiogenic factors, such as VEGF-A and bFGF, as well as biochemical agents, including heparin, chymase, and tryptase. This neovascularization supplies the necessary oxygen to hypoxic areas, stabilizing the lesion and reducing the accumulation of dead cells, which in turn diminishes plaque formation [204].

However, angiogenesis can have negative consequences, including tissue damage through the formation of extracellular DNA traps (NETs). MCs have been associated with several cardiovascular events, including acute myocardial infarction, angina pectoris, and cardiomyopathies, characterized by an increase in cardiomyocytes and vascular cells. They also play a protective role in myocardial infarction by regulating myofilament calcium sensitivity and post-infarction cardiac function [205]. In Cpa3Cre/+ mice, MCs deficiency was associated with decreased cardiomyocyte contractility by releasing tryptase, which activates protease-activated receptor 2 (PAR-2) on cardiomyocytes [206]. However, both W/Ww Kit knockout and Cpa3Cre mice may have undetected alterations that impact study results, including mutations in other cells expressing the same receptor, the absence of certain cell types, or changes in gene expression [207]. Moreover, DNA insertion into the genome can disrupt the function of adjacent genes, leading to unexpected effects such as gene deletion or alterations in other biological processes [208].

This activation promotes the release of angiogenic factors, such as VEGF-A, and lymphangiogenic factors, including VEGF-C, which aid in resolving cardiac dysfunction through the process of lymphangiogenesis. Evidence highlights the multifaceted role of MCs in the cardiovascular system [209]. Specifically, studies on stenotic human aortic valves reveal a correlation between aortic stenosis severity and the co-localization of macrophages, DCs, and lymphocytes, along with fibrosis and MCs density in the leaflets. In hearts with valvular heart failure, concentrations of histamine, tryptase, and mast cell density are elevated compared to healthy controls [210]. The presence of MCs-secreting granules may act as an autocrine factor, promoting MCs hyperplasia in cardiac tissue and contributing to collagen accumulation through the local release of fibrotic factors, such as histamine and leukotriene C4. However, the severity of the condition may vary depending on MC levels [211].

Numerous studies highlight the significant role of vasoactive and inflammatory mediators released by MCs in cardiovascular health, impacting the cardiac environment [212]. The excessive release of these compounds can cause detrimental changes in cardiovascular function, leading to alterations in blood vessel structure and modifications to cardiac tissue, which may result in pathological conditions [213]. The increase in MCs in cardiovascular diseases warrants attention, as their proliferation has both positive and negative effects. While MCs defend against infections and injuries, their excessive increase can escalate inflammation, aggravating cardiac tissue damage, and facilitating the progression of conditions such as atherosclerosis. In addition, the role of IL-10 secreted by MCs, which modulates inflammation, adds complexity to their function [214].

In summary, the interplay of mediators, inflammation, and MC activity has a significant influence on cardiovascular health, with both beneficial and detrimental implications. However, much of the evidence for mast cell activity remains speculative [215]. Key questions arise: How do mediators from MCs impact the cardiac environment? What are the specific benefits or harms associated with increased MC populations in cardiovascular disease? Which subsets of cardiac mast cells are dangerous to cardiac health, and if the population changes during cardiovascular disease, what harm or benefit does the heart gain from the change? How and in what way might mast cells play a protective role in pathological conditions?

## 6. Controversies

The role of MCs in diseases such as cancer, cardiovascular disorders, and inflammatory conditions remains controversial, largely due to limited rigorous studies addressing their specific contributions. Although essential to immune function, their roles in various diseases are complex and often debated, underscoring the need for more detailed investigations.

### 6.1. Heterogeneity in the Activity of MCs in Cancer

Cancer-associated MCs exhibit wide discordance, with studies reporting both pro-tumorigenic and anti-tumorigenic effects. This dichotomy stems partly from methodological heterogeneity across studies and the limitations of target-specific experimental models. Pro-tumorigenic mechanisms: MCs promote cancer progression by releasing histamine, proinflammatory cytokines, and angiogenic factors [216]. These effects are mediated via *c-kit*/stem cell signaling and histamine receptors (H1/H2). In contrast to their pro-tumor functions, mast cells possess anti-tumorigenic capabilities through the secretion of mediators that promote apoptosis and recruit cytotoxic immune cells [217,218]. A paradigmatic example is non-small cell lung cancer (NSCLC), where MCs exhibit dual roles, modulating tumor growth through molecules such as SCF, FGF-2, IL-8, VEGF, PDGF, and NGF. They also express anti-tumor mediators (e.g., TGF-β, TNF-α, proteases, and IL-10). Notably, some of these molecules have context-dependent effects, exerting either pro- or anti-tumorigenic activity [219]. A relevant example is the involvement of intestinal MCs in the progression of colorectal cancer (CRC). Studies in murine models (C57BL/6 and BALB/c strains) have explored the immunological microenvironment of CRC, but their findings have limitations. The mutation in *c-Kit* affects not only MCs, but also basophils and intestinal γ/δ T cells, which distorts the interpretation of their specific role.

Furthermore, in mice with APC mutations that develop spontaneous intestinal tumors, significant differences are observed between tumors of the small intestine and colon [220]. These findings underscore the need to address key aspects, such as Advanced functional studies in tumor initiation and immune cell manipulation, as well as the analysis of MC heterogeneity, including the identification of progenitors that generate subpopulations with opposing functions (e.g., antitumorigenic vs. protumorigenic MCs). Characterization of the role of cytokines and factors secreted by MCs in CRC progression. To date, research has generated more questions than certainties about the exact role of MCs in this context [221].

A promising line of research could focus on VEGF molecules secreted by MCs, as the data currently available are largely from tumors subjected to conditions of malnutrition and oxygen deprivation (hypoxia). These conditions favor the release of proangiogenic signals by cancer cells, which drives the expansion of the tumor vascular network. However, it has not been demonstrated, and it is only suspected that MCs perform this task. Additionally, preliminary studies suggest that histamine may play a role in the growth of tumors. If so, treatment with antihistamines could theoretically inhibit this process; however, there is as yet no conclusive research in this regard. This approach would not only be relevant for non-small cell lung cancer (NSCLC), but also for other types of cancer, such as colorectal and breast cancer. Some other products, such as antibodies (imatinib), have been shown to increase prostate tumor growth, leading to the inference that MCs play a role in tumor progression, both as tumor protectors and destroyers [222]. A promising area of research could focus on the role of VEGF molecules secreted by MCs. Current data primarily come from tumors exposed to conditions of malnutrition and oxygen deprivation (hypoxia), which tend to enhance the release of pro-angiogenic signals by cancer cells. This process contributes to the expansion of the tumor vascular network. However, it remains unproven, and it is only suspected that MCs play a role in this mechanism.

Additionally, preliminary studies suggest that histamine may play a role in the growth of tumors. If this is the case, treatment with antihistamines could potentially inhibit this process; however, conclusive research on this idea is still lacking. This approach could be relevant not only for NSCLC but also for other types of cancer, including colorectal and breast cancer. Moreover, certain products, such as the antibody imatinib, have been shown to increase prostate tumor growth, suggesting that MCs play a dual role in tumor progression, acting both as protectors and destroyers [223].

Evidently, due to the lack of specific studies, it has long been assumed that, since MCs are components of the tumor microenvironment and promote cell growth in other contexts, they would also do so in cancer. Initially, they were associated with stimulating neoangiogenesis, tissue remodeling, and modulating the immune response. Nevertheless, this modulation was extrapolated to the cancer context without solid evidence, inferring that, by secreting proteases and given that some cancers are activated through receptors stimulated by proteases, mitogen-activated kinases, prostaglandins, and histamine, this would favor tumor progression by the action of MCs. In addition, several molecules (such as bFGF, transforming growth factor β (TGF-β), TNF-α, IL-8, metalloproteinases (MMPs), tryptase, and chymase) have been linked to mechanisms that inhibit apoptosis, thereby allowing cancer cell survival and reducing cell motility. Mediators of oese processes include chymase, tryptase, TNF-α, IL-1, and IL-6. These assumptions have taken for granted the role of MCs in cancer. However, this approach has delimited potential therapeutic targets, shortening the path for future targeted research [223].

Finally, several articles explore the influence of the tumor microenvironment, although most are review studies that analyze the release of molecules within this setting. Similar to non-targeted research, these works primarily establish associations between processes in one context and potential outcomes in another, without offering direct evidence. For instance, while it is hypothesized that malignant cells actively release microparticles, facilitating the transfer of membrane receptors and regulatory molecules (e.g., microRNAs) to tumor and immune cells, the specific effects of these molecules on tumors and MCs remain unexplored experimentally.

Furthermore, greater emphasis should be placed on human studies, as reliance on mouse models remains prevalent. This approach raises a critical question: Do mast cells truly exhibit identical characteristics in humans and mice? A fundamental concern is whether murine models accurately replicate human diseases, particularly since these animals are often genetically modified to mimic human pathologies, a practice that casts doubt on the translational validity of the findings. Another layer of controversy arises from the functional overlap between MCs and other immune cells, including T cells and DCs. In some cases, it is assumed that the roles of certain cells could be compensated for by others, despite a lack of direct evidence implicating MCs. This assumption presents a significant challenge: treatments effective for pathologies involving T cells or DCs may not necessarily yield the same results in MC-mediated tumors [224].

### 6.2. The Debated Role of MCs in Cardiovascular Pathologies

The function of MCs in cardiovascular pathologies is complex and dualistic, as evidence suggests they can contribute to both tissue repair and disease aggravation, making their overall role controversial. Although MCs have been present in the heart, aorta, and other vessels since ontogeny, producing unique mediators and expressing distinctive markers, their presence does not automatically confirm their pathogenic involvement [201]. Its increase has been associated with cardiac fibrosis, but its mechanisms of action remain unknown. Studies in humans and animal models show an increase in MCs in various heart diseases, such as hypertension, acute myocardial infarction, valvular stenosis, myocarditis, cardiomegaly due to volume overload, heart failure (especially with arrhythmias), and transplanted hearts [225]. Cardiac MCs would modulate the extracellular matrix (MEC), promoting MEC repair or activating degradation, with an excessive presence of type I and type III collagen, the latter being a key mechanism in cardiac fibrosis, a common finding in these pathologies [226]. It has been known for years that cardiac fibrosis involves myofibroblasts and the MCs (with release of hormones, growth factors, cytokines, and integrins) as a source of fibrosis by their proteases. However, there is an important gap in the understanding of the participation and function of resident cardiac MCs and in the identification of the stimuli for their active participation that allow them to be promoters of a profibrotic environment [227]. An increased density of MCs is associated with fibrotic regions in endomyocardial biopsies from patients with heart disease. In addition, an increase in the number of MCs has been observed in the right ventricle of rats subjected to pulmonary artery constriction, a model that induces right ventricular fibrosis. Similarly, in transplanted human hearts, the number of MCs also increased and correlated with the degree of fibrosis. Clinical findings corroborate this relationship: patients with a higher number of MCs at 2 weeks after transplantation developed more fibrosis at week 3. Likewise, in those with a lower probability of recovery after acute myocardial infarction, the number of MCs presented a significant increase [228]. There are more studies on the intervention of MCs in cardiovascular pathologies than in other pathologies. Regarding the controversy on the role of MCs in heart disease, we consider that the scientific evidence already available in many articles should be adequately evaluated.

### 6.3. MCs in Inflammatory Diseases: Conflicting Inflammatory or Anti-Inflammatory Mechanisms

The role of mast cells in immunity includes the potential production of nitric oxide (NO), a crucial signaling molecule. However, controversy exists as to whether MCs produce NO, due to the paucity of conclusive evidence. Although some studies suggest that MCs produce NO via nitric oxide synthase (NOS), this claim has not been consistently demonstrated [229,230]. Likewise, the involvement of mast cells in liver diseases is a matter of debate, given the complexity of their role in the pathogenesis of these conditions. While they are known to be involved in some inflammatory processes and the immune response, the evidence for their specific role in liver damage is heterogeneous and requires further investigation. Since, as in the previous cases, some add up to benefits, while others add up to harm. Some studies highlight benefits at the liver level, while other research focuses on the excessive activation of MCs and the release of mediators such as histamine and tryptase, which can trigger inflammation, fibrosis, and ultimately, chronic liver damage. Moreover, their accumulation can lead to their dysfunction [231,232,233].

MCs appear to have a dual role in inflammatory diseases, as they can both promote inflammation and participate in its control and eventual elimination. The latter is recognized in the repair attributed to the release of IL-10. However, its dysregulation could contribute to the development of various chronic inflammatory and autoimmune diseases. This association is due to the release of inflammatory mediators, including proinflammatory and chemotactic molecules, which attract other immune cells to the site of inflammation or damage, thereby perpetuating a redundant cycle of inflammation. MCs have been implicated in diseases such as rheumatoid arthritis, inflammatory bowel disease, asthma, psoriasis, and acne, where their excessive activation promotes chronic inflammation and tissue damage. They also play a role in systemic lupus erythematosus and multiple sclerosis, contributing to both exacerbation of the inflammatory response and tissue deterioration [234].

There is substantial evidence that MCs play an important role in autoimmune diseases. However, it is generally believed that they function primarily as accessory cells in autoreactive autoimmune responses, which are driven primarily by antibodies or T cells. Research on mast cells has been carried out in animal models of autoimmune diseases, although these results have some limitations. Moreover, the interactions involved are so complex that signaling networks and cellular behaviors remain unclear, making it difficult to draw definitive conclusions from isolated observations, but their importance should not be underestimated [235]. Several authors attribute to MCs the primary role in autoimmune diseases, as they can exacerbate the pathology by acting as mediators (with a TNF-like release mechanism) and/or by suppressing the function of regulatory T cells (Tregs). However, this relationship has only been demonstrated in murine models of rheumatoid arthritis and multiple sclerosis. Among the experimental strategies to elucidate its highlights, the use of an Fcε-Fcγ fusion protein, designed to bind with high affinity to the human FcγRIIb receptor, which prevents the activation and degranulation of MCs, allows for the evaluation of its involvement in the disease. Likewise, masitinib, a selective tyrosine kinase inhibitor, has demonstrated efficacy in reducing the survival, migration, and activity of MCs, and has already been evaluated with promising results in multiple sclerosis [236]. Another of the most controversial debates within inflammatory processes is to understand the difference between MCAD and MSDA. Identification is complex due to the absence of reliable biomarkers for diagnosis, which is based solely on syndromic criteria. The relevant and controversial aspects of MCs in the liver are important because they are found in large quantities, distributed in the portocaval system, hepatic arteries and veins, and bile ducts, independently of the presence of lesions, and increase in cases of liver damage. The regulatory role of MCs has been demonstrated in inflammatory diseases, where they modulate the responses of T lymphocytes and Kupffer cells [237,238].

An example of the contradictory in hepatic MCs is that they secrete mediators critical for eosinophil differentiation, chemotaxis, and activation, such as IL-3, IL-5, GM-CSF, and PAF, which may benefit the host. Mast cells in the liver are closely associated with the innervation of the sympathetic and parasympathetic nervous systems, suggesting that they may influence or be influenced by nerve fibers. Some authors postulate that nerves stimulate MCs, increasing the release of fibrotic factors in patients with primary biliary cholangitis (PBC), primary sclerosing cholangitis, bile duct obstruction, hepatitis, alcohol-induced liver injury, steatosis, steatohepatitis, and other diseases. It suggests that mast cells play an active role in PBC and other liver diseases. The authors found a significant increase in the number of *c-kit*-positive, chymase- and tryptase-positive MCs located near nerve fibers. Research on the potential intervention of MCs in other inflammatory pathologies, such as acne, scleroderma, and psoriasis, among others, is still in its early stages of development. Therefore, it is necessary to address this topic in greater depth and detail in future work dedicated exclusively to these diseases [239].

## 7. Conclusions

It is highly unlikely that data on embryogenic hematopoiesis will be obtained in humans, since direct and massive tests of cell differentiation in this line cannot be performed in humans. This is because the process probably begins in the aorta and the yolk sac, then moves to the liver, and finally to the bone marrow, making it difficult to study. The organs where it develops are essential for life, which is why the data obtained, allowing us to transfer it to the human scenario, comes from species such as mice, where most studies on fetal hematopoietic development, signaling, receptor expression, and others have been conducted. In the study of mast cell receptors and their behavior, more studies have also been conducted in mice, especially those involving a mutation pathway. The first studies on the use of tyrosine kinase blockers were also conducted in mice [240].

This review highlights the complex and dualistic nature of mast cells (MCs), positioning them as multifunctional immune sentinels with roles that far exceed their classical association with allergic diseases. MCs originate from hematopoietic progenitors, with their development and tissue-specific maturation governed by a precise network of transcription factors and cytokines. Their remarkable plasticity enables their phenotypic and functional characteristics to be shaped by the local tissue microenvironment, allowing for tailored responses to diverse physiological and pathological contexts.

MCs are now recognized as crucial players in maintaining tissue integrity and homeostasis. They are indispensable in processes such as wound healing, tissue remodeling, angiogenesis, UV radiation protection, and reproductive physiology. Furthermore, as first responders, they serve a vital protective role in host defense against a wide array of pathogens, including bacteria, viruses, parasites, and fungi, through both IgE-dependent and independent mechanisms.

Conversely, the dysregulation of MCs is central to the pathogenesis of numerous diseases, including allergic diseases driven by IgE-mediated activation, as well as a spectrum of Mast Cell Activation Disorders (MCADs), such as Mastocytosis, Mast Cell Activation Syndrome (MCAS), and Hereditary Alpha-Tryptasemia (HAT), all characterized by inappropriate mediator release. Their role is also critical in complex conditions such as Chronic Spontaneous Urticaria (CSU) and cardiovascular diseases, where they exhibit a controversial dual role, contributing to pathologies like atherosclerosis while also potentially engaging in protective repair mechanisms.

Therapeutic advancements directly targeting MC pathways represent a significant breakthrough. Novel therapies, including anti-IgE monoclonal antibodies (e.g., Omalizumab), tyrosine kinase inhibitors (e.g., imatinib, avapritinib), BTK inhibitors (e.g., Ibrutinib), and anti-KIT antibodies (e.g., barzolvolimab), offer promising strategies for treating conditions such as severe asthma, chronic urticaria, and systemic mastocytosis.

In the case of cancer, the primary strategies involve the use of inhibitors that block the action of abnormal proteins (imatinib, masitinib, sunitinib) that send signals to cells promoting tumor development and the pro-tumor effect of mast cells. The inhibition specifically targets protein kinases. MCADs are currently being managed with antibody-based treatments and tyrosine kinase inhibitors [169]. Given the ongoing controversy regarding MC involvement in these contexts, further targeted studies are warranted.

Despite this progress, the role of MCs remains contentious in several areas, highlighting critical knowledge gaps. Their function in cancer is paradoxical, displaying both pro- and anti-tumorigenic effects that require clarification in human-specific models. Their precise contribution to cardiovascular repair versus damage remains debated, and their primary involvement in inflammatory diseases, such as psoriasis and liver pathologies, requires stronger validation. Future research must prioritize human studies over murine models and employ advanced functional analyses to resolve these controversies, unravel the full spectrum of MC heterogeneity, and identify novel, precise therapeutic targets. In summary, mast cells are versatile master regulators of immunity and tissue homeostasis; understanding their nuanced functions is crucial for developing effective treatments for a wide range of diseases.

## Figures and Tables

**Figure 1 ijms-26-08895-f001:**
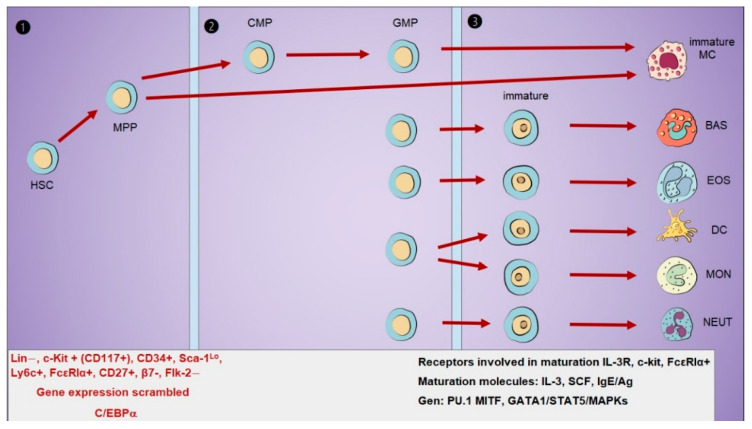
The evolutionary line of MCs from bone marrow to maturation. ❶ Hematopoietic Stem Cell Progenitors (HSC), Multipotent Progenitors (MPPs), and mast cells can be generated from the MPPs lineage directly or by the GMP lineage. ❷ Common Myeloid Progenitors (CMPs), Granulocyte/Monocyte Progenitors (GMP). ❸ Immature granulocytes and monocytes from the GMP precursor lineage. It has also been proposed that MPPs in mice give rise to immature MCs and cells that remain in the heart from embryogenesis onwards, generating mast cells throughout life (red: bone marrow; black: housing site).

**Figure 2 ijms-26-08895-f002:**
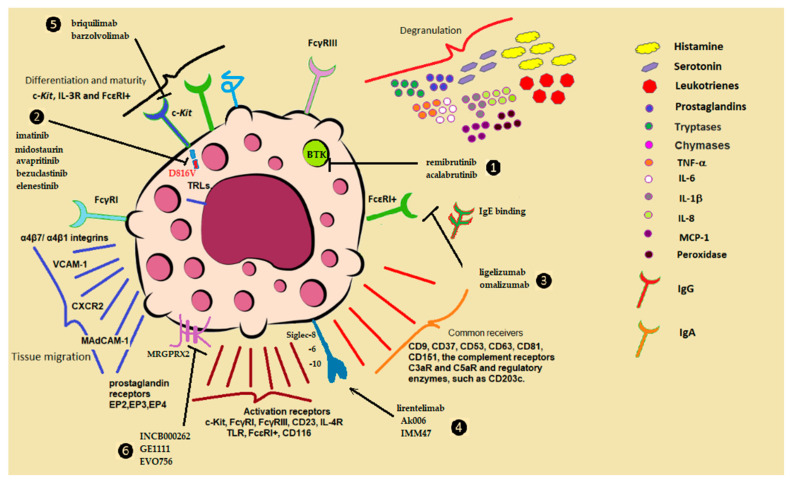
The receptors expressed in the bone marrow within the MPP lineage include Lin−, c-Kit+ (CD117+), CD34+, Sca-1Lo, Ly6c+, FcεRIα+, CD27+, β7−, and Flk-2. These receptors play a role in migration and maturation: SCFR, c-Kit, CD117, IL-3 receptor (CD123) (purple key; MC activation), interleukin-4 receptor (IL-4R), granulocyte-macrophage colony-stimulating factor receptor (GM-CSFR, CD116), and CD27 (all receptors are black keys). For tissue migration, the relevant receptors include integrins α4β7/α4β1, CXCR2, MAdCAM-1, VCAM-1, and CCR2/CCL-2. In the intestine, M2 integrin and glycoprotein IIb. Prostaglandin receptors: EP2, EP3, and EP4, α3- and α5-laminin, TRLs (royal blue key). Major cellular receptors of mature MCs are: tetraspan family; CD9, CD37, CD53, CD63, CD81, CD151, complement receptors C3aR and C5aR and regulatory enzymes, such as CD203c (orange key). Chemokines, cytokines, and degranulation molecules, including VEGF and interleukins IL-6 and IL-8, which affect IL-4 and other cytokines. Degranulation: red key molecules. Therapy targeting mast cells, receptors, and signaling pathways: ❶ BTK (Bruton’s tyrosine kinase) inhibitor, which is being developed to treat chronic spontaneous urticaria (CSU). ❷ FLT3 receptor signal transduction inhibitors in the treatment of systemic mastocytosis, and *Kit*-D816V mutation, an inhibitor of tyrosine kinases. ❸ Monoclonal IgG1 specific for IgE. ❹ Siglec, sialic acid-binding immunoglobulin-type lectin with humanized monoclonal antibody. ❺ Humanized antibodies inhibiting c-Kit activation by SCF. ❻ binding to and inhibiting the activity of tryptase, an enzyme found in MCs. Drugs targeting mast cells can act directly, as is the case with imatinib, midostaurin, and avapritinib, which inhibit tyrosine kinases in receptors such as c-Kit. In the case of omalizumab and ligelizumab, they bind to free circulating IgE and block its action on target cells. Remibrutinib dysregulates genes in inflammatory signaling pathways and immunoglobulin production.

**Figure 3 ijms-26-08895-f003:**
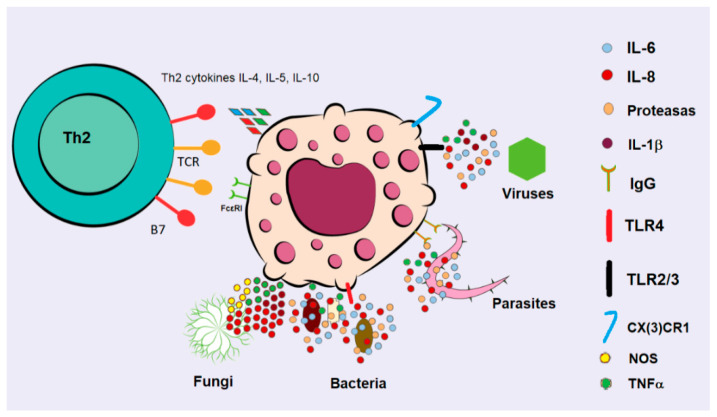
Receptors and mediators play a crucial role in controlling infections. In Th2 cells, receptors (TCR) are activated via FcεRI or CX3CR1, binding to the TCR of lymphocytes and leading to the secretion of Th2 cytokines, such as IL-4, IL-5, and IL-10, in response to parasites. Cytokines such as TNF-α, IL-6, and IL-8 are secreted in the presence of fungi through the activation of Toll-like receptors (TLRs), C-type lectin receptors (CLRs), and Pathogen-Associated Molecular Patterns (PAMPs). FcγRIIα, also present in MCs, can be activated by IgG molecules, causing degranulation. Bacteria and viruses can be opsonized by IgGs, which activate cytokine secretion, or be identified through the TLR4 present in MCs, which detects lipopolysaccharides in bacterial cell walls. This interaction stimulates the release of IL-6, IL-8, and proteases. When they encounter parasites, MCs recognize IgE-mediated opsonization with their FcεRI receptor and release proteases and TNF-α. TLR2/3, IgG, and IgM are activated in viral and parasitic infections, which initiates complement activation, nitric oxide synthase (NOS) production, and the release of proinflammatory cytokines.

**Table 1 ijms-26-08895-t001:** Direct activation by non-IgE mediators.

Mediator	Activation Mode	Response	References
IgG, adenosin.	β-hexosaminidase (β-hex) Lysosomal enzyme that degrades GM2 gangliosides, a group of nerve tissue chemicals.	IgG binds to high-affinity FcγRI in autoimmune diseases, neuropathies, and certain types of cancer.	[56]
Sphingosine-1-phosphate (S1P) Lysophosphatidic acid (LPA).	Lipid that, through its S1PR1-S1PR5 receptors, which are G protein-coupled receptors on MCs.	Contact hypersensitivity reaction for IgLC.	[57]
Free light chain immunoglobulins (IgLC).	IgLC and IgE bind specifically to neurons and mast cells at FcεRIα.	It rapidly releases 5-HT, H1, proteases, LT4, TNF-α, and MIP-2 (vasodilation and permeability). Promotes antigen-MHC to the late-phase response. Increases hypersensitivity, thereby exacerbating asthma, food allergies, and other conditions.	[58,59,60]
Substance P (SP) Neuropeptide.	Activates MCs when bound to G protein (MRGPRX2).	Its activation affects multiple sclerosis and psoriasis, neurodegenerative diseases, and infections.	[61,62]
Stem Cell Factor (SCF).	Receptor c-kit (CD117).	A potent stimulant for MC proliferation, differentiation, and survival.	[63,64]
Interleukin 3 (IL-3).	IL-3 and IL3Rα (CD123) receptor.	Together with SCF, FcεRI, crucial for MCs differentiation and maturation.	[12,65,66,67]
Interleukin 4 (IL-4). Interleukin 13 (IL-13).	Activated MC, Th2 cells, basophils, and eosinophils produce IL-4. IL-4 and IL-13 genes are closely related.	Inhibits production of IL-1, TNF-α, IL-6, and MIP-1β. Potently inhibits apoptosis, differentiation, and proliferation of B lymphocytes. Promotes asthma, dermatitis, and anaphylaxis. The allergic model involves IL-4 and IL-13 acting on the vasculature, sensitizing it to histamine, platelet-activating factor (PAF), or leukotriene C4 (LTC4).	[68,69,70]
Interleukin 9 (IL-9).	IL-4, IL-33, and TGFβ induce the production of IL-9.	IL-9 exhibits antiparasitic activity and exacerbates the IL-9- and MC-dependent allergic response.	[71,72,73,74]
Other cytokines.	IL-33, IL-5, IL-6, IL-10, TSLP, and GM-CSF.	Enhance human mast cell and basophil proliferation, both stem cell factor (SCF)-dependent and IgE/antigen-independent.	[75,76,77]
	MC cytokines: IL-1, IL-1β, and IL-18.	Promote Th2 cytokine production and promote the expansion and differentiation of APCs, dendritic cells (DCs), macrophages, and MCs.	[78,79]
Complement and anaphylatoxins C3a-C5a.	Through their receptors, C3aR and C5aR. C1q for pathogen opsonization.	Induces degranulation and chemotaxis in human MCs (C1q in cutaneous MCs and C3a for pulmonary MCs).	[80,81,82]
Neurotensin (NT) and the Corticotropin-releasing hormone (CRH).	Modulate MC activity.	It responds to stress, inflammation, and nervousness and is secreted locally at nerve endings. CRH also induces FcεRI receptor expression.	[83,84,85,86]
Neurotrophins: Nerve growth factor (NGF), Brain-derived neurotrophic factor (BDNF), Neurotrophin-3 (NT-3) and Neurotrophin-4/5 (NT-4/5).	MC-dependent regulation of neuronal survival and function. NTs are secreted mainly by neurons and glial cells.	Biological effects are mediated through two types of receptors: p75 and Trk. Highlights the significant role of neurotrophins in the development and survival of MCs. NT-3 promotes fetal MCs maturation.	[85,87,88,89]
Pattern recognition receptors (PRRs). Pathogen-Associated Molecular Patterns (PAMPs). Damage-Associated Molecular Patterns (DAMPs).	Viral particles, parasites, and various endogenous molecules activate toll-like receptors in MCs. TLRs: 2, 4, 5, 9.	Induces degranulation and chemotaxis of MCs.	[90,91,92,93]

**Table 2 ijms-26-08895-t002:** Mast Cell Dysfunction and Disease.

Mast Cell Dysfunction and Disease (MCAD) Is a General Term for When MCs Release Mediators Inappropriately, Causing Disease.		
Rankings of Mast Cell Activation Syndrome (MCAS)		
MCAS: Mast Cell Activation Syndrome		
**Primary MCAS**	**Secondary MCAS**	**Idiopathic MCAS**
(1) Low threshold for desgranulation. (2) Increase in MC population, with increased response.	(1) More common and of unclear etiology. (2) IgE-mediated mechanisms (an environmental allergen, such as food or medication) and non-IgE-mediated mechanisms (such as exercise).	(1) There are no positive medical, laboratory, or diagnostic data. (2) There are no allergic causes or clonal mast cell diseases.
Diagnostic Criteria		
**A: clinical criterion**	**B: laboratory criterion**	**C response criterion**
Mastocytosis		
**Cutaneous mastocytosis**	**Systemic mastocytosis**	**Mast cell sarcoma and Leukemia**
(1) Maculopapular cutaneous mastocytosis. (2) Diffuse cutaneous mastocytosis.	Systemic mastocytosis in bone marrow. Slow-progressing systemic mastocytosis. Systemic mastocytosis with neoplasia. Aggressive systemic mastocytosis. Mast cell leukemia.	(1) 30% of biopsies (+) with serum tryptase and/or D816V mutation of the Kit gene. (2) Myelodysplastic signs, hepatomegaly, acyrthosis, and others.
Hereditary Alpha-Tryptasemia Is a Genetic Condition		
**Genetic cause**: Duplications or extra copies of the TPSAB1 gene, which is responsible for producing tryptase.	**Autosomal dominant trait:** Which means it can be inherited from one or both parents.	**Elevated tryptase levels**: Extra copies of the gene result in increased production of the tryptase protein, which is detectable in blood tests and can lead to anaphylaxis.
**Symptoms:** People with hereditary alpha-tryptasemia may experience a range of symptoms, including: Flushing. Itching (pruritus). Gastrointestinal problems, such as dysmotility. Autonomic nervous system dysfunction. Anaphylactic or anaphylactoid reactions.		

**Table 3 ijms-26-08895-t003:** The WHO Classification of Mastocytosis.

WHO Categories	Subtypes	Diagnostic Criteria	Characteristics
Cutaneous Mastocytosis.	Urticaria pigmentosa. Diffuse cutaneous mastocytosis.	Immunohistochemistry.	Reddish papular lesions. Reddish diffuse thickening of skin.
	Solitary mastocytoma of skin.		Brownish-yellow, minimally elevated plaque.
Systemic mastocytosis.	Indolent SM.	Diagnostic criteria of SM.	No C findings At least 2 B findings and no C findings.
	Aggressive SM.		At least one C finding.
	SM is associated with clonal hematological non-mast cell lineage disease.		WHO criteria for a clonal hematological neoplasm.
Mast cell leukemia.		Diagnostic criteria of SM.	No C findings. At least 2 B findings, and no C findings. Isolated BM mastocytosis.
Mast cell sarcoma.		No criteria for SM.	PB smear > 10% MCs. Isolated MC tumor. Destructive growth pattern.
Extracutaneous Mastocytoma.		No criteria for SM.	Isolated MC tumor. Non-destructive growth pattern.

BM: Bone marrow, MCs: mast cells, PB: peripheral blood, SM: systemic mastocytosis, WHO: World Health Organization.

**Table 4 ijms-26-08895-t004:** Activation of mast cells in the CSU.

Allergy Is Mediated by IgE and FcεRI Receptors.	Widely Recognized for the Recognition.	Specific Allergen.
IgE/FcεRI cross-linking molecules.	IgG, IgM, free IgG.	High allergen variability.
Autoallergy or autoreactive urticaria type I or autoimmune urticaria and autoimmune urticaria IIIb.	IgE vs. Auto-antigens. More than 200 different self-antigens were not detected in healthy controls.	The most frequent are IL-24, DNA, and Thyroid Peroxidase (TPO) IgG anti-TPO.
Complement.	Infections.	Autoantibodies activate C1 (classical). Alternative cascade, by C3a cleaving C5 to form C5a, and in the coagulation cascade, thrombin (FIIa) and factor Xa.
Coagulation.	Coagulation factors IIa (thrombin), VIIa and Xa are ligand-receptor 2 (PAR2) on the mast cell surface.	Activated by C5a, TLR-4 can be activated by fibrin.
Neuro-immune dysregulation.	Receptor-2 (PAR2) ligands interact with the Mas-related G protein-coupled X2 receptor (MRG). X2 (MRGPRX2) is expressed on MCs.	Susntacia P, Neurotrophins, Neurotensins, etc.
Leukotrienes.	Product of constitutively and de novo formed granules following mast cell activation (LTC4-LTD4).	Which activation product and LTB4 can be chemotactic for immature MCs.
Alarmins.	PRRs, PAMP, DAMP, Stem Cell Factor; ST2 (*c-kit*).	This IL-33 alarmin is released as an alarm signal in case of infection or epithelial damage.
T cells, CD4^+ +^.	Receptor; TLR ligand, CD40 ligand.	TPO, thyroid peroxidase, OX40 Receptor, TCR, Licos.
Infections.	PRRs.	Toll-like receptors.

## Data Availability

Not applicable.

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
