# Peer review of "Role of Mast Cells in Human Health and Disease: Controversies and Novel Therapies"

_ijms, 2025, doi:10.3390/ijms26188895_

Round 1
Reviewer 1 Report
Comments and Suggestions for Authors
Title: Role of Mast Cells in Human Health and Disease: Controversies and Novel Therapies.
In this paper, the authors investigate the role of mast cells in immunity, focusing on their contributions to homeostasis, cancer, and various other diseases. They conclude that this research aims to clarify the primary functions of mast cells by examining the molecules they release or those present in their environment, as well as their roles.
Abstract: The abstract should be reformulated, eliminating old references (Von Recklinghausen in 1863 and Paul Ehrlich in 1877…), which could be inserted into the introduction. Furthermore, the conclusion does not address the essence of the paper; it should provide something new.
This article seems good to me; however, I have some concerns.
The figures are only mentioned but not described in their meaning in the legend.
If the authors have data on human mast cells, this chapter should be developed, since the existing works in the literature only concern murine mast cells.
The article is very long and many paragraphs that deal with obsolete concepts should be removed.
The discussion and conclusion must focus on new data.
A description of pro- and anti-inflammatory cytokines is missing.
One of the main intracellular signalling pathways activated in mast cells is the MAPK pathway. Activation of MAPK in mast cells leads to degranulation and the production of cytokines and chemokines. In this regard, I suggest that the authors briefly mention this topic. Below, I insert a new important article that should be studied, incorporate the meaning, and report it in the references.
Saggini R, Pellegrino R. MAPK is implicated in sepsis, immunity, and inflammation. International Journal of Infection. 2024;8(3):100-104. (www.biolife-publisher.it)
For better presentation and completeness of this article, the authors should briefly mention IL-4, which increases IgE production by B lymphocytes and fuels the allergic circuitry. In mastocytosis, excess IL-4 amplifies inflammation and allergic symptoms. Additionally, elevated levels of IL-31 are observed in patients with mastocytosis, which correlate with the severity of itching. Again, in this regard, I suggest that the authors briefly mention this topic. Below, I insert a new important article that should be studied, incorporate the meaning, and report it in the references.
Tei M. Role of IL-4 and IL-31 in mastocytosis. International Journal of Infection. 2024;8(1):18-19. (www.biolife-publisher.it)
I believe that these suggestions help to complete the picture of this paper. Therefore, without these corrections, the article cannot be published.
I would like to revise the paper after the authors' corrections.
My suggestion is "Minor Revision".
Comments on the Quality of English LanguageThe English could be improved
Author Response
Thank you very much for taking the time to review this manuscript.
Point-by-point response to comments and suggestions for authors and in the manuscript with track changes.
Reviewer Comments
In this paper, the authors investigate the role of mast cells in immunity, focusing on their contributions to homeostasis, cancer, and various other diseases. They conclude that this research aims to clarify the primary functions of mast cells by examining the molecules they release or those present in their environment, as well as their roles.
Specific comments,
Comments 1. Abstract: The abstract should be reformulated, eliminating old references (Von Recklinghausen in 1863 and Paul Ehrlich in 1877…), which could be inserted into the introduction. Furthermore, the conclusion does not address the essence of the paper; it should provide something new.
Response 1: Rewrote the summary, line 15.
Response 1: Thank you for your comment; the abstract has been reworded. (Von Recklinghausen in 1863 and Paul Ehrlich in 1877....) It has been moved to lines 36-39 of section 1.1. Identification of mast cell precursors.
Response 1: The conclusion was reworded, section 6.
Comments 2. The three figures already describe the content of the image.
Response 2: The three figures already describe the content of the image and are described.
Comments 3. If the authors have data on human mast cells, this chapter should be developed, since the existing works in the literature only concern murine mast cells.
Response 3: No, there are only tests on mice, as stated in the footnote to Figure 1.
Comments 4. The article is very long, and many paragraphs that deal with obsolete concepts should be removed.
Response 4: The paragraphs containing obsolete concepts have been removed from the article, thereby reducing its length.
Comments 5. A description of pro- and anti-inflammatory cytokines is missing.
Response 5: We have included a paragraph on pro-inflammatory and anti-inflammatory molecules in section 1.3 Receptors for maturation and tissue migration, where many cytokines are mentioned. Line 174, page 4.
Comments 6. One of the main intracellular signalling pathways activated in mast cells is the MAPK pathway. Activation of MAPK in mast cells leads to degranulation and the production of cytokines and chemokines. In this regard, I suggest that the authors briefly mention this topic. Below, I insert a new important article that should be studied, incorporate the meaning, and report it in the references.
Saggini R, Pellegrino R. MAPK is implicated in sepsis, immunity, and inflammation. International Journal of Infection. 2024;8(3):100-104. (www.biolife-publisher.it)
Response 6: A paragraph from the information contained in the article by Raoul Saggini (2024) is included. In section 2.1. IgE/antigen-dependent activation. Line 204, page 5.
Comments 7. For better presentation and completeness of this article, the authors should briefly mention IL-4, which increases IgE production by B lymphocytes and fuels the allergic circuitry. In mastocytosis, excess IL-4 amplifies inflammation and allergic symptoms. Additionally, elevated levels of IL-31 are observed in patients with mastocytosis, which correlate with the severity of itching. Again, in this regard, I suggest that the authors briefly mention this topic. Below, I insert a new important article that should be studied, incorporate the meaning, and report it in the references.
Response 7: The role of IL-4 and IL-31 in urticaria and atopic dermatitis, the provocation of increased IgE was included. It was included in section 4.4 Urticaria, line 551, page 15.
Reviewer 2 Report
Comments and Suggestions for Authors
A comprehensive overview of mast cell biology. Usefull for practicing allergologists.
General remarks: there are some contradictions in the description of clinical syndromes (specified in Specific remarks), some names of the drug are written capitalised. In particular, there is a bit confusion among sections Mastocytosis, Mast cell activation syndrome (MCAS) and Mast cell activation disorders (MCAD). Could you explain the classification of those clinical syndromes in a single table?
Specific remarks
Line 41: Mast cells (MC) are distributed in all body tissues s) are found in all body tissues, PROBABLY MISTAKE, TWICE THE SAME STATEMENT
Line 377-8: FcεRI can also be activated by other molecules, including IgG, which triggers degranulation. CAN YOU PLEASE EXPLAIN, HOW IgG ACTIVATES FcεRI
Lines 398: It should be mentioned that majority of patients with indolent forms of mastocitosis are diagnosed after presenting with severe anaphylaxis to Hymenoptera venoms.
Line 401: Elenestinib --> elenestinib
Line 423: The cited article is a rewiev article. Please cite the research where the prevalence ob 17% was established.
Line 424-6 As I know the prevalence of cKIT gain in mutation in general population is fass below 1%. The statement that the c-Kit mutation, especially the KIT D816V mutation, serves as a consistent marker for the syndrome is in contrast with the poposed prevalence of disease among pregnant women. The reference 172 only compares the sensitivity of cKIT detection form blood compared with a bone marrow biopsy, and the article 173 is a PubMed analysis of classification of MCAS.
Line 431-2: (2) secondary MCAS, induced by an underlying IgE-dependent allergies or immunological disorders. Please make a comment, if a patient with let's say allergic rhinitis falls under umbrella of MCAS.
I suggest to include a simple and recent classification system like this one: Current Allergy and Asthma Reports (2024) 24:39–51 https://doi.org/10.1007/s11882-024-01126-0
Line 465-6: A very important message (Additionally, MCs can release mediators selectively, enhancing the secretion of other neuro-responsive mediators [185]). However, there is no reference to support that fact. The cited article is a review article.
Line 467: The current treatment is based on either omalizumab or Ligelizumab In fact, the current treatment is based on allergen avoidance and mast cell mediator antagonists. Ligelizumab should not be capitalised.
Line 483: I miss omalizumab as the mostly used biologic drug for chronic urticaria.
Line 492: Remibrutinib should not be capitalised.
Lines 510-1: In addition, Siglec-6 is not expressed at very low levels than in CMs. The sentence is strange, I don't understand the meaning.
Line 515: the sentence Acute urticaria is associated with the release of inflammatory mediators. Probably doesn’t fit in that paragraph, but should be moved to line 521.
Line 521: Acute urticaria typically involves an IgE-mediated type 1 hypersensitivity In fact some cases of acute urticaria are type 1 hypersensitivity, however majority of patients with acute urticaria have different pathogenesis, mostly acute infection.
Line 531: some names of drugs are capitalised.
Lines 537 and following: The section MCAD is very confusing. Please make this paragraph consistent with previous paragraphs of macrocytosis and MCAS. There is a new abbreviation CM, which is not explained
Author Response
Thank you very much for taking the time to review this manuscript.
Point-by-point response to comments and suggestions for authors and in the manuscript with track changes.
Reviewer Comments
General remarks: there are some contradictions in the description of clinical syndromes (specified in Specific remarks), some names of the drug are written capitalised. In particular, there is a bit confusion among sections Mastocytosis, Mast cell activation syndrome (MCAS) and Mast cell activation disorders (MCAD). Could you explain the classification of those clinical syndromes in a single table?
Response: The names of the medications have already been rewritten with lowercase letters. In the Mastocytosis section, a table was created to clarify the difference between Mast Cell Activation Syndrome (MCAS) and Mast Cell Activation Disorders (MCAD), line 404.
Specific comments,
Comments 1. Line 41: Mast cells (MC) are distributed in all body tissues s) are found in all body tissues. PROBABLY MISTAKE, TWICE THE SAME STATEMENT
Response 1: Now, Line 40: already corrected.
Comments 2. Line 377-8: FcεRI can also be activated by other molecules, including IgG, which triggers degranulation. CAN YOU PLEASE EXPLAIN, HOW IgG ACTIVATES FcεRI.
Response 2: Now, lines 390 contain an error: it is not FcεRI but FcγRIIα. The receptor has already been replaced.
Comments 3. Lines 398: It should be mentioned that majority of patients with indolent forms of mastocitosis are diagnosed after presenting with severe anaphylaxis to Hymenoptera venoms.
Response 3: A suggested reference with citation has been included in line 398... now 416.
Comments 4. Line 401: Elenestinib --> elenestinib
Response 4: Line 401 has been corrected: Elenestinib --> elenestinib and all names of the drug, now line 427. And all drugs in the manuscript should be lowercase (those that are not trademarks).
Comments 5. Line 423: The cited article is a review article. Please cite the research where the prevalence of 17% was established.
Response 5: Citation 171 has been replaced by the one from the study where they obtain 17%, now citation and line 448.
Comments 6. Line 424-6 As I know the prevalence of cKIT gain in mutation in general population is fass below 1%. The statement that the c-Kit mutation, especially the KIT D816V mutation, serves as a consistent marker for the syndrome is in contrast with the poposed prevalence of disease among pregnant women. The reference 172 only compares the sensitivity of cKIT detection form blood compared with a bone marrow biopsy, and the article 173 is a PubMed analysis of classification of MCAS.
Response 6: We left the first article (172, now 177) unchanged, maintaining its specificity for use in suspected cases of patients treated by obstetrician-gynecologists, and modified the second article (173, now 178).
Comments 7. Line 431-2: (2) secondary MCAS, induced by an underlying IgE-dependent allergies or immunological disorders. Please make a comment, if a patient with let's say allergic rhinitis falls under umbrella of MCAS.
Response 7: No, not unless you develop mast cell activation syndrome. That paragraph was deleted and as suggested a table (Table 2) was made with the recent classification.
Comments 8. I suggest to include a simple and recent classification system like this one: Current Allergy and Asthma Reports (2024) 24:39–51 https://doi.org/10.1007/s11882-024-01126-0
Response 8: The suggested changes, the classification of MCAS and diagnosis of MCAS were made. Now on page 12, line 406.
Comments 9. Line 465-6: A very important message (Additionally, MCs can release mediators selectively, enhancing the secretion of other neuro-responsive mediators [185]). However, there is no reference to support that fact. The cited article is a review article.
Response 9: The appropriate backup reference was placed ref 185, now 188.
Comments 10. Line 467: The current treatment is based on either omalizumab or Ligelizumab In fact, the current treatment is based on allergen avoidance and mast cell mediator antagonists. Ligelizumab should not be capitalised.
Response 10: Thanks for the information. All medications listed in the manuscript are already written in lowercase.
Comments 11. Line 483: I miss omalizumab as the mostly used biologic drug for chronic urticaria.
Response 11: Omalizumab was cited for urticaria, now line 502.
Comments 12. Line 492: Remibrutinib should not be capitalised.
Response 12: It has been corrected.
Comments 13. Lines 510-1: In addition, Siglec-6 is not expressed at very low levels than in CMs. The sentence is strange, I don't understand the meaning.
Response 13: The paragraph and citation were removed.
Comments 14. Line 515: the sentence Acute urticaria is associated with the release of inflammatory mediators. Probably doesn’t fit in that paragraph, but should be moved to line 521.
Response 14: It has been corrected.
Comments 15. Line 521: Acute urticaria typically involves an IgE-mediated type 1 hypersensitivity In fact some cases of acute urticaria are type 1 hypersensitivity, however majority of patients with acute urticaria have different pathogenesis, mostly acute infection.
Response 15: Thanks for the information
Comments 16. Line 531: some names of drugs are capitalised.
Response 16: Yes, at the beginning of a sentence and registered trade names. Barzolvolimab (CDX-0159) and briquilimab are not written with capital letters. Thank you very much.
Comments 17. Lines 537 and following: The section MCAD is very confusing. Please make this paragraph consistent with previous paragraphs of macrocytosis and MCAS. There is a new abbreviation CM, which is not explained.
Response 17: The section has been deleted.
Round 2
Reviewer 2 Report
Comments and Suggestions for Authors
The review comments are sufficiently improved.